# The Underestimated Role of the p53 Pathway in Renal Cancer

**DOI:** 10.3390/cancers14235733

**Published:** 2022-11-22

**Authors:** Alessandra Amendolare, Flaviana Marzano, Vittoria Petruzzella, Rosa Anna Vacca, Luisa Guerrini, Graziano Pesole, Elisabetta Sbisà, Apollonia Tullo

**Affiliations:** 1Department of Biosciences, Biotechnologies and Environment, University of Bari Aldo Moro, 70121 Bari, Italy; 2Institute of Biomembranes, Bioenergetics and Molecular Biotechnologies, National Research Council—CNR, 70126 Bari, Italy; 3Department of Translational Biomedicine and Neuroscience (DiBraiN), University of Bari Aldo Moro, 70121 Bari, Italy; 4Department of Biosciences, Università degli Studi di Milano, 20133 Milan, Italy; 5Institute of Biomedical Technologies, National Research Council—CNR, 70126 Bari, Italy

**Keywords:** p53, RCC, p53 pathway, mutated TP53

## Abstract

**Simple Summary:**

The TP53 tumor suppressor gene, the guardian of the genome, is mutated or has an inactive pathway in most chemo-resistant tumors. Most studies focus on the role of mutated TP53 in tumors. In this review, we discuss the role of p53 pathway alterations in renal cell carcinoma, one of the most chemo-resistant tumors that require clinical and resolutive approaches.

**Abstract:**

The TP53 tumor suppressor gene is known as the guardian of the genome, playing a pivotal role in controlling genome integrity, and its functions are lost in more than 50% of human tumors due to somatic mutations. This percentage rises to 90% if mutations and alterations in the genes that code for regulators of p53 stability and activity are taken into account. Renal cell carcinoma (RCC) is a clear example of cancer that despite having a wild-type p53 shows poor prognosis because of the high rate of resistance to radiotherapy or chemotherapy, which leads to recurrence, metastasis and death. Remarkably, the fact that p53 is poorly mutated does not mean that it is functionally active, and increasing experimental evidences have demonstrated this. Therefore, RCC represents an extraordinary example of the importance of p53 pathway alterations in therapy resistance. The search for novel molecular biomarkers involved in the pathways that regulate altered p53 in RCC is mandatory for improving early diagnosis, evaluating the prognosis and developing novel potential therapeutic targets for better RCC treatment.

## 1. Principal Features of Renal Cell Carcinoma

Renal cell carcinoma (RCC) represents a class of heterogeneous genitourinary tumors that rank among the ten most common tumors worldwide. Indeed, its incidence is increasing especially in recent years, with a worldwide rate of relapse and mortality of over 40%, representing the sixth most frequently diagnosed cancer in men and tenth among women [1,2,3]. Through recently identified morphological and molecular criteria, RCCs have been classified into three main different histopathological subtypes: clear cell RCC (ccRCC), the most common renal malignancy (~70–80% of all RCC); papillary RCC (pRCC) (~10–15% of all RCC); and chromophobe RCC (chRCC) (~5–10% of all RCC). Moreover, there are two other subtypes: collecting duct RCC (cdRCC, ~1%) and sarcomatoid RCC (srRCC) (≤1%) [4,5]. They are all very aggressive tumors, and up to 17% of patients harbor distant metastases already at the time of diagnosis [6]; after the complete surgical resection of the primary tumor, there is a recurrence in about 30% of patients [7,8]. Tumor recurrence is mainly due to RCC resistance to radiotherapy and chemotherapy [8], and despite new therapeutic approaches such as immunotherapy, immune cell cycle checkpoint inhibitors (ICIs) and therapeutic vaccines, a significant percentage of patients with renal carcinomas are not benefiting from these therapies. Moreover, cancer treatment and diagnosis based on metabolites are still in the early stages and require further development for possible clinical application [9]. Therefore, understanding the molecular pathways underlying tumorigenesis in the early stages is needed to define novel molecular biomarkers that can be used for early diagnosis and evaluate the prognosis of RCC and develop novel potential therapeutic targets.

Several potentially relevant biomarkers for RCC are reported in the literature, such as Von Hippel-Lindau (VHL), survivin, XIAP (X-linked inhibitor of apoptosis), MCL-1 (myeloid cell leukemia-1), HIF1α (hypoxia-inducible factor 1α), HIF2α, NRF2 (nuclear factor erythroid 2-Related Factor 2), transglutaminase 2 (TGase 2), MDM2 (mouse double minute 2), TP53/p53, KRAS, and AKT [10], but for many of them, there are conflicting data on whether they play a role as oncogenes or tumor suppressors and whether they may be considered effective biomarkers.

Among the proposed biomarkers, a crucial role is certainly played by p53 since the cellular activities of most biomarkers identified in RCC are in some way dependent on or related to p53. For example, the crosstalk between p53 and tumor suppressor gene VHL is crucial in the DNA-damage response mediated by p53 in RCC [11]. Mutations in the VHL gene have been the first identified genetic alterations leading to RCC [4,12]. Mutated VHL results in the upregulation of HIF-1α. Indeed, in normoxic condition, hydroxylation by prolyl hydroxylases (PHDs) of HIF-1α in its oxygen-dependent degradation domain (ODD) is required for the interaction of HIF-1α with VHL, leading to the ubiquitination and proteasomal degradation of HIF-1α. When the cells are hypoxic, PHDs are inactivated, this leads to HIF-1α stabilization and translocation to the nucleus, where it forms a dimer with HIF-1β and binds and promotes the expression of genes involved in a series of cellular responses, including survival, cell death, metabolic reprogramming, angiogenesis, stemness, inflammation, metastasis, immune evasion, etc., that is, genes that help tumor cells to adapt to the hypoxic environment and contribute to tumor progression [13,14]. Meanwhile, p53 responds to several cellular stresses including hypoxia, and the loss of p53 functions and hypoxia are two common events in cancer progression, indicating a close but complex interplay between p53 and HIF1α [15]. Indeed, some studies reported that HIF-1α induces MDM2 inhibition, leading to p53 stabilization and the activation of apoptosis [16]. On the contrary, other studies reported that hypoxia and HIF-1α negatively regulate p53 stability and activities in some cell lines [15]. It is observed that in some tissues, severe hypoxia increases p53 stability and activity, leading to cell death, while mild hypoxia decreases p53 activity, promoting cell survival. Moreover, it is well known that MDM2 is a p53 target gene and is the main negative regulator of p53 stability, mediating its direct ubiquitination for proteasome-mediated degradation to maintain p53 protein at a low level under non-stressed conditions. Moreover, MDM2 polymorphism is an independent adverse prognostic factor for RCC, and MDM2 upregulation is associated with an increased risk of developing RCC. It has been reported that MDM2 may play additional roles beyond regulating p53 stability. Indeed, it has been demonstrated that it is involved in the regulation of HIF1α. For instance, it has been reported that in the HCT116 colorectal cancer cell line, p53 acts as a scaffold protein to bridge MDM2 to HIF-1α, leading to the ubiquitination and proteasomal degradation of HIF-1α by MDM2 [17,18]. Nevertheless, other studies showed that MDM2 can increase the levels of HIF-1α [18,19], suggesting that the effect of MDM2 on HIF- 1α could vary depending on the cell/tissue type and severity of hypoxia [10,15].

Remarkably, the fact that p53 is poorly mutated in RCC does not imply that it is functionally active, and increasing experimental discoveries have demonstrated this. In particular, ccRCC is a clear example of cancer in which despite wild-type p53 status, a poor response to conventional anti-cancer treatments is occurring, thus ccRCC represents an extraordinary example of the importance of p53 pathway alterations in therapy resistance.

In this review we will report the role of p53 and the main genes involved in its pathways in renal carcinomas, and intend to prove that the proper functioning of p53 is the basis of effective chemotherapy and radiotherapy in anti-cancer treatments.

## 2. p53, the Main “Gatekeeper” of the Genome

The p53 protein is considered one of the main regulators of the cell cycle as it is stabilized and activated following different cellular stresses, such as ionizing radiation (e.g., γ-rays), electromagnetic radiation (e.g., ultraviolet radiation), biological stresses (viral infections or bacterial toxins), chemical stresses, (toxic substances), endogenous stresses (high production of ROS, reactive oxygen species, which are highly damaging to all cellular structures), hypoxia or uncontrolled activation of oncogenes [20]. In response to such DNA-damaging events, p53 is capable of triggering several biological responses, such as cell cycle arrest, DNA repair, senescence, apoptosis, autophagy, ferroptosis, DNA repair, metabolism adaptation, cell migration/invasion, modulation of oxidative stress, so much so as to earn the name of “the guardian of the genome”. Its importance in the cell is witnessed by the fact that functional inactivation of the p53 protein has been found in about 90% of human cancers. In approximately 50% of the cases, this inactivation is due to point mutations in the TP53 gene [21,22,23], while the remaining 40% depends on the alteration of p53 positive or negative modulators.

The TP53 gene architecture includes 11 exons and 10 introns and encodes for an extremely preserved sequence-specific transcription factor of 53 kDa, composed of 393 amino acids with seven functional domains [24,25,26]. Two transactivation domains (TAD-1 and TAD-2) are present in the N-terminal portion, followed by a proline-rich domain (PRD), a DNA-binding domain (DBD), and a hinge domain (HD). In the carboxyl terminus of the protein, the oligomerization domain (OD) and a negative regulation domain (α), both rich in lysines are present (Figure 1). The latter domain together with the TADs domains undergo several posttranslational modifications (phosphorylation, acetylation, methylation, ubiquitinylation, sumoylation, neddylation, etc.) that regulate p53 activity and stability [27].

p53 controls a variety of cellular functions acting as a transcription factor. It can bind in a sequence-specific manner the regulatory regions of more than 300 protein-coding genes, but also many non-coding genes, including microRNAs and long non-coding RNAs, regulating either positively or negatively their expression. In each of them, there is a sequence recognized and bound by p53, termed p53RE (Responsive Element). The p53RE consists of a double decamer with sequence RRRCWWGYYY (where R = A/G, W = A/T, Y = C/T) separated by a spacer sequence between 0 and 13bp [28]; this sequence is bound by a p53 tetramer, composed of two p53 dimers, each of which binds one half of the consensus sequence [29].

In a healthy cell, the p53 level is kept low by MDM2-E3 ubiquitin ligase mediated ubiquitination and subsequent degradation. In response to different stress stimuli, p53 phosphorylation prevents the MDM2-p53 interaction, and hence p53 is stabilized and regulates the expression of its target genes.

Alongside tumor suppression function, p53 plays important functions in other biological and pathological processes, such as metabolic diseases, aging, ischemia, neurodegeneration, tissue injuries immune response, viral infection, maternal reproduction and development [22,30,31,32,33,34,35].

### 2.1. Regulation of p53 Stability and Activity

In normal conditions, p53 half-life is of about 30 min; in stress conditions, p53 half-life is enhanced up to 2/3 h by post-translational modifications that can occur on p53 or on MDM2, impairing the p53/MDM2 interaction. The main post-translational modifications are phosphorylation, acetylation, methylation and ubiquitination. Phosphorylation is a crucial post-translational event in the regulation of p53 stability; it occurs in all functional domains and it can be mediated by a wide range of protein kinases, including ATM (ataxia telangiectasia mutated), ATR (ataxia telangiectasia and Rad3-related protein), DNA-PK (DNA-dependent protein kinase), Chk1, Chk2 (checkpoint kinase-1/Checkpoint kinase-2) and JNK (c-Jun N-terminal kinase). Depending on the phosphorylated residue, p53 triggers different cellular responses ranging from cell cycle arrest to DNA repair and apoptosis regulating a different subset of target genes in each condition [36].

p53 acetylation mainly concerns the C-terminal lysine residues, catalyzed by acetyltransferases as CBP (p300/CREB-binding protein), PCAF (p300/CBP-associated factor), TIP60 (60 kDa Taf Interactive Protein) and MOF [37,38,39]. Acetylation of the Lys120 residue results in p53-dependent induced apoptotis; simultaneous acetylation of Lys120 and Lys164 residues, contributes instead to cell cycle arrest. In some cases, acetylation and ubiquitination occur on the same lysines; consequently, these are competitive and mutually exclusive events [40].

p53 ubiquitination and degradation occurs when the cell is in a physiological state or has re-established a physiological state following stress events. p53 ubiquitination is mainly promoted by the E3 ubiquitin-ligase MDM2 (Mouse Double Minute 2) protein. The interaction between p53 and MDM2 constitutes a self-regulatory cycle with p53 transcriptionally activating the MDM2 gene; MDM2 requires two cofactors, UBE4B and WIP1, to regulate p53 half-life [41,42]; on the other hand, low levels of MDM2 cause p53 mono-ubiquitination with subsequent translocation to mitochondria and loss of its nuclear transcription activity [43]. However, MDM2 is not the only E3 ubiquitin ligase that regulates p53 levels, in fact when MDM2 is inhibited, p53 can still be ubiquitinated and degraded [44] by the E3-ligases MDMX (MDM4), COP1, Pirh2 or Arf-BP1 [45].

MDM2 can also modify p53 by mediating the transfer of several UBiquitin-Like proteins (UBL) as SUMO-1, SUMO-2/3 and NEDD8. p53 sumoylation can be triggered also by specific SUMO-E3-ligase like Topor, PIAS and several TRIM proteins. Some of these E3-ligase-mediated modifications are associated with p53 inhibition while others are involved in the regulation of transcriptional activity, cell cycle control and subcellular transport. An example is the ubiquitin-E3-ligase E4F1, which promotes p53 oligo-ubiquitination on lysine residues (K319–K321), distinct from those targeted by MDM2, resulting in the p53-dependent transcriptional activation of target genes involved in growth arrest [46].

A molecule that counteracts MDM2 activity and thus acts as an inducer of p53 stability is the nucleolar protein ARF (Alternate Reading Frame), which can bind MDM2, preventing its interaction with p53 and subsequent degradation [47]. In normal cells ARF has a low steady state that dramatically increases when cells are under stress conditions [48]. Furthermore, ARF can inactivate also other p53 negative modulators, such as Arf-BP1 [49].

### 2.2. p53 Isoforms

For a long time, the TP53 gene was thought to generate a single transcript, but 20 years from its discovery multiple p53 isoforms have been described [25]. The use of alternative promoters, splicing events and alternative translational start sites allow the generation of twelve p53 isoforms (Figure 1). When transcription starts from the P1 promoter, located upstream of exon 1, three different p53 isoforms are produced: p53α is the longest isoform since the corresponding mRNA contains all 11 exons; this protein contains at the N-terminus the trans-activation domains TAD 1 and 2, the oligomerisation domain (OD), the DNA-binding domain (DBD) and the C-terminal domain; this is the predominant p53 isoform, the only one studied form in more than 20 years.

p53β and p53γ include exons 1 to 9, plus the additional exon 9b present in intron 9. The presence of an additional AUG codon at position 40 (alternative translation start site) or the alternative splicing event of intron 2 lead to the formation of a category of protein isoforms, the ∆40-p53, lacking the first 39 amino acids and thus lacking the TAD1 domain, but still retaining the second one and the entire DNA binding domain; similarly to what has been described above, in the case of ∆40-p53 there are α, β, γ isoforms, as a consequence of alternative splicing at 3′ (Figure 1) [25]. These isoforms have a reduced ability to activate the transcription of p53 target genes, but they can form complexes with other p53 isoforms and positively or negatively modulate p53-dependent gene expression depending on their relative levels and cellular context.

If, on the other hand, transcription is controlled by the P2 promoter located in intron 4, the ∆133-p53 (α, β, γ) isoforms, lacking the first 132 amino acids at the N Terminal, are generated; these isoforms lack TAD domains 1 and 2 and part of the DBD domain (Figure 1) [50]. Δ133p53 lacking the N-terminal region that mediates the interaction with MDM2, escapes proteasomal degradation, but not the autophagic one [51]. Furthermore, Δ133p53 forms heterocomplex with p53 and consequently modulates gene expression in a p53-independent way.

Finally, the presence of a third AUG codon at position 160 leads to the expression of the ∆160-p53 (α, β, γ) isoforms truncated of the first 159 amino acids [25]. These isoforms have lost both transactivating domains and a part of the DNA binding domain but can still heterodimerized with the other isoforms [52].

Each isoform seems to be involved in distinct functions, being able to modulate p53 activities and to have an impact on clinical parameters [53,54]. p53β and p53γ have been shown to promote apoptosis in the breast cancer MCF-7 cell line [55], and p53β can enhance the transactivating activity on p21 and Bax promoters, while p53γ can stimulate the transactivating activity of the Bax promoter exclusively [55]. Moreover, high levels of p53β are positively associated with better prognosis in breast cancer and acute myeloid leukemia patients [56,57].

Δ40p53, which lacks the N-terminal domain corresponding to the MDM2 binding site on p53, can heterodimerize with p53 and induce or inhibit p53 transcriptional activity depending on the cellular context [54].

As for the ∆133p53 isoform, it has been reported to be involved both in cell activities promoting cell survival, angiogenesis and metastasis as well as in cellular senescence and apoptosis [58,59].

It seems that ∆160p53 is expressed mainly in cells with mutp53 promoting tumorigenesis [60]. It has also been shown that in melanoma patients higher levels of ∆160p53 are associated with more aggressive tumors [61].

## 3. Wild-Type and Mutant p53 in RCC

There is no doubt that mutations in the p53 gene are always related to the development of a tumor with an unfavorable diagnosis. Although mutations in about 190 different codons are reported in the literature, the eight most common ones are found in the DNA-binding domain: R175H, Y202C, G245S, R248Q, R248W, R273C, R273H, and R282W (Figure 1). The primary outcome of TP53 mutation is the loss of functions of wild-type p53 and the acquisition of new deleterious functions that support cell proliferation (gain of function mutations). Indeed, mutant p53 (mutp53) in heterozygosity loses the ability to transactivate its target genes and exerts a dominant trans-repressive effect on its wild-type counterpart, sustaining cell proliferation, cell migration, metastasis and chemoresistance that transform it from the guardian of the genome into the guardian of cancer cells [62].

The rate of TP53 gene mutations in renal tumors is surprisingly low, especially in ccRCC. Fengzhi Li et al. performed a somatic p53 mutation analysis in the three major types of RCCs. They found that the mutation rate of TP53 is higher in chRCC with a frequency of 31.8%, while TP53 in ccRCC and pRCC has a much lower mutation rate at 3.24% and 2.48%, respectively. Moreover, TP53 mutations are clearly associated with poor patient survival in all three major RCC types [10]. The codons mutated in a higher percentage are found in the DNA-binding domain and encode for Arginine at positions R175, R248, and R273, although mutations have also been observed in the oligomerisation domain particularly in position R337, specifically in chRCC, and R342, which appears almost exclusively in Wilms’ tumor that affects children [63]. The mutation patterns are different for different histopathological subtypes of RCC. In chRCC, the highest rate of mutations (frameshifts, missense and nonsense mutations) is observed at position R213. In ccRCC the highest frequency of mutations are in G244, R273, P278, K132 and C135 codons [14].

However, so far there is only a small amount of research focusing on the analysis of the p53 status in patients affected by RCC, and therefore there is not enough data to provide a complete picture of the TP53 mutation pattern and function when it comes to this particular cancer. Moreover, it has recently been reported that RCC patients who have mutations in the TP53 gene and in the SMARCA4 (BRG1) gene, which is part of the SWI/SNF remodeling complex, have a poor prognosis [64,65].

### 3.1. p53 Isoforms Expression in Renal RCC

There are still conflicting results on the role of different p53 isoforms in carcinogenesis and tumor progression. However, it seems that p53 isoforms are deregulated in different cancers, including RCC, and might participate in p53 inactivation, in tumor initiation and progression.

An analysis of 45 RCC patients with different tumor stages showed that the expression of almost all p53 isoforms changes during cancer development and progression [66]. In the early stages of carcinogenesis, an increase in mRNA levels of p53β and p53γ is observed that in later stages decreases returning comparable to non-tumor tissues. Interestingly, another study reports that patients with a high level of p53β, exhibit improved recurrence-free survival (RFS) and overall survival compared with those who have a low level of p53β isoforms even if they have a mutated p53 [67]. Indeed, the overexpression of p53β induces increased apoptosis with increased expression of Bax and Caspase-3 regardless of p53 status, suggesting p53β as an important indicator of better prognosis for patients [67].

In more advanced stages of carcinogenesis, the ∆40p53 and ∆40p53γ isoforms appear to increase. Furthermore, the up-regulation of the ∆40p53 isoform has been observed in RCC patients with mutated p53, although no association with patient survival was observed [68].

However, cells from advanced-stage tumors with and without mutations in the TP53 gene show a different expression pattern of p53 isoforms that varies with treatment with Topotecan (a topoisomerase 1 inhibitor), but they do not represent a predictor of response to treatment with this chemotherapeutic agent [66].

Marijana Knezovic Florijan et al. analyzed 41 RCC tissues and found that the expression of p53, Δ40p53 and Δ133p53 was upregulated in RCC with mutp53 compared with RCC tissues with wtp53, but there was no difference in the expression of these isoforms compared with normal adjacent tissues. This study underlines the importance of considering both the expression of p53 isoforms and the mutational status of p53 in RCC clinical studies [68].

In contrast to these studies, Diesing and colleagues analyzed a cohort of 55 RCC patients and found no associations between different p53 isoforms’ expression, clinical features or advanced tumor stages. Unexpectedly, they only observed an association between the expression of the ∆133p53α isoform, which promotes cell survival and metastasis, and smaller tumor size [69,70,71,72].

In conclusion, p53 and its isoforms are key factors in mediating cell response and cell sensitivity to treatment in RCC, though more studies are required to better understand the role of network interactions between p53 and its isoforms in this cancer.

### 3.2. p53 Inactivation in RCC

The fact that in renal carcinomas p53 is wild-type does not necessarily mean that it is functional since mutations and expression alterations in proteins that regulate p53 stability and activity are another way to render p53 functionally inactive.

It has been reported that VHL protein is required to fully activate p53 functions, and the perturbation of the interplay between p53 and VHL seems to explain the resistance of RCCs to chemotherapy [14]. Moreover, in many tumors that have mutations in the VHL gene, the expression of the proteins involved in p53 stability and activity is deregulated, and as a consequence, p53 is not functional despite its wild-type status, suggesting that VHL and p53 activities are somehow dependent on each other in the cell. Since VHL has been shown to bind p53 through its α-domain, it has been hypothesized that VHL stabilizes p53 by facilitating its phosphorylation, thereby preventing its binding to MDM2, its main negative regulator [11,73]. This would explain why VHL-mutated cells are resistant to therapies despite harboring wild-type p53. Indeed, in these cells, the exogenous expression of wild-type VHL makes them sensitive to adriamycin or sunitinib, resulting in significantly increased cell proliferation arrest and apoptosis.

Another interesting trigger that can lead to p53 and VHL inactivation in RCCs is the overexpression of some miRNAs. Indeed, the overexpression of the miR-17-92 cluster was reported in ccRCC and in other urological tumors, including Wilms tumor and bladder and testicular cancers [74]. This cluster functions as an oncogene in collaboration with the MYC oncogene, which promotes its expression. The interesting thing is that many genes with a key role in tumorigenesis are targets of the miR-17-92 cluster miRNAs, such as VHL, TRIM8, PTEN, EGFR, mTOR, PI3K and VEGF [74,75,76]. Therefore, the overexpression of these miRNAs in ccRCC may be another mechanism leading to p53/VHL inactivation.

Other authors have speculated that the lack of p53 functionality in RCC cells could be due to the hypoxic condition that is generated in a rapidly growing tumor. Indeed, hypoxia leads to the accumulation of p53 but also of HIF-1α, which has competitive activities to those of p53 [69].

Another reason why p53 could be not functional in RCC cells could be linked to the downregulation of p53BER2 RNA (RNA-p53 bound enhancer region 2), which is an enhancer RNA (eRNA) involved in promoting efficient p53 transcription activity [77].

Fengzhi Li et al. conducted a comparison between MDM2 expression in RCC tumors and normal tissues and found that in chRCC, MDM2 was significantly decreased in early stage 1 of the cancer compared with normal tissues while significantly increased in early stage 1 in both ccRCC and pRCC [10]. That is, in ccRCC and pRCC cancer cells, MDM2 could act as an oncogene for its role in wild-type p53 degradation. Moreover, a link between MDM2 and HIF1α transcription factor was proved because the siRNA-mediated downregulation of MDM2 decreased the expression of HIF1α and HIF2α in VHL-defective RCCs and increased VEGF and PAI-1 [78].

Recently, TRIM8 protein, a 61.5 kDa E3 ubiquitin-ligase, was found to be a p53 modulator. It is a member of the TRIM (tripartite motif) family defined by the presence of a common domain structure composed of a tripartite motif including a RING-finger, one or two B-box domains and a coiled-coil motif [79]. Alterations of TRIM expression levels represent a biomarker and prognostic factor of specific cancers [80]. It was demonstrated that TRIM8 is a direct p53 target gene and that following a genotoxic stress, p53 transactivates TRIM8, which through a positive feedback loop displaces MDM2-p53 binding, thus stabilizing p53, which in turn triggers the transcription of cell cycle arrest genes (p21) and DNA repair genes (GADD45) [81] (Figure 2).

A prominent role for TRIM8 in regulating cancer cell growth was shown in vivo in ccRCC; the TRIM8 expression level was significantly decreased in tumors compared with matched non-tumor tissues, and this signature is typical of more malignant neoplasms since it was not found in benign oncocytomas (RO). The TRIM8 downregulation observed in ccRCC patients is due to the up-regulation of the miR-17-5p and miR-106b-5p, whose overexpression is promoted by the oncogene MYC-N [74] (Figure 2). It has been demonstrated that miR-17-5p and miR-106b-5p directly target the 3′UTR of TRIM8 and repress its expression as well as other tumor suppressors, such as p21 and PTEN. Thus, deficits in TRIM8 impair p53 activity because p53 cannot be stabilized by TRIM8. In turn, p53 cannot transactivate the expression of genes such as GADD45 and p21 involved in DNA repair and cell cycle arrest, respectively. Interestingly, the silencing of miR-17-5p and/or miR-106-5p by specific anti-miRNAs leads to the recovery of the p53 stability and activity mediated by TRIM8 and a decrease in MYCN expression mediated by miR34a, whose expression is activated by p53. Consequently, the proliferation of ccRCC cells decreased, and their sensitivity to chemotherapeutic drugs increased [75,82] (Figure 2).

Moreover, in ccRCC cell lines, TRIM8 promotes the degradation of the oncogenic isoform ΔNp63α both in a caspase 1-dependent manner and proteasomal way, but only in a functional p53 background [83].

### 3.3. p53 Role in Multi-Drug Resistance in RCC

Tumor relapse after drug treatment represents a very critical problem for clinicians and patients. This is due to the features of cancer cells of already being or becoming resistant to chemotherapeutic agents. Indeed, chemoresistance can be intrinsic, or primary, if the tumor cells harbour mutations in genes that confer resistance to therapy from the beginning before the administration of the drug and acquired, or secondary, if the tumor cells are initially sensitive to the chemotherapeutic agents and only after therapy develop resistance due to the acquisition of new genetic mutations by a subset of malignant cells that makes them more aggressive [84]. When these tumor cells become resistant to a wide range of drugs, they develop multiple drug resistance (MDR). It is widely accepted that all tumors have an intratumoral heterogeneity, consisting also of nonmalignant cells such as fibroblasts, immune cells and other structures such as extracellular vesicles (EVs) and extracellular matrix (ECM), which greatly contributed to the response to chemotherapy and in the selection of cellular subclones resistant to treatment [85].

DNA damage and replication are the main targets of most chemotherapeutic agents. Therefore, the main chemoresistance mechanisms are due to the impaired transmembrane transport of the drugs within cells (mainly the downregulation of the afflux SLC proteins [solute carrier transporter] and the up-regulation of the efflux ABC proteins [ATP-binding cassette]) or to an inability of the cells to respond properly to drug-induced damage, such as DNA damage repair, or properly execute a programmed cell death [86].

Once again, p53 plays a key role in the regulation of intracellular and extracellular microenviroment pathways involved an effective responses to chemotherapy. Indeed, p53 is also involved in the regulation of cellular secretome, which modulates microenvironmental parameters such as ECM, vascularization and pH as well as intercellular communication and interactions that can affect the behavior of neighboring cells [87,88]. Consequently, it is not by chance that p53 mutations or inactivation are present in most chemo-resistant tumors. Moreover, mutant TP53 turns the tumor suppression function of wild-type p53 into tumor promotion by acquiring new functions that are very important in determining or developing the chemoresistance of cancer cells.

Indeed, p53 regulates many factors responsible for resistance to cisplatin, one of the most widely used chemotherapeutic agent in clinical oncology. One of these is the Nuclear Factor erythroid 2-related factor 2 (Nrf2) which is downregulated by wild-type p53 and upregulated by mutated p53 [89]. Indeed, it was found that the mutated p53 tunes Nrf2-dependent antioxidant response, promoting tumor cell survival, and the p53-Nrf2 axis upregulates the proteasome machinery, conferring resistance to proteasome inhibitors used in cancer therapy.

Mutated TP53, in particular the R248Q mutation, contributes to the resistance of tumors to doxorubicin, a chemotherapy drug that inhibits DNA replication and transcription by blocking the activity of topoisomerase II and upregulating ABCB1 transporter [86]. Moreover, mutated TP53 at His175 or at amino acid residues 22 and 23 in the N-terminal domain resulted in resistance to fluoropyrimidine antimetabolite 5-Fluorouracil (5-FU), another chemotherapy drug widely used in clinical oncology that blocks the activity of thymidylate synthase [86].

In general, for the treatment of renal carcinomas, most chemotherapeutic agents, including platinum compounds (cisplatin, oxaliplatin, carboplatin), anthracyclines (e.g., doxorubicin), anti-metabolites (e.g., 5-fluorouracil, methotrexate and gemcitabine), vinca alkaloids (vincristine, vinblastine and vinorelbine) and taxane compounds (docetaxel, paclitaxel) have failed to be clinically useful [90,91]. Kidney cancers express high levels of ABC proteins, although inhibitors of ABC carriers were unsuccessful in improving chemotherapeutic outcomes [92,93,94,95]. Immunotherapy also did not give the desired results, while kinase inhibitors such as sorafenib and sunitinib greatly improved the treatment of advanced kidney cancer [96,97,98]. Recently, the drug Temsirolimus was approved, an inhibitor of mTOR (mammalian target of rapamycin), which represents an alternative for those patients who did not respond to kinase inhibitors [99,100].

However, despite all recent efforts to make chemotherapy treatments in kidney cancers more effective, RCC cancers still remain fatal in a large percentage of patients if the tumor is not removed in the early stages of development. Therefore, it is crucial to understand the mechanisms that determine the chemoresistance in RCC, in particular the mechanisms that lead to the p53 inactivation, since TP53 is poorly mutated in RCC as previously illustrated.

Recently, RLIP76, a novel glutathione-electrophile conjugate (GS-E) and multidrug transporter, has been proposed as a new promising target [101]. RLIP76 expression correlates with the degree of tumor malignancy and predicts decreased survival in cancer patients. Its over-expression suppressed apoptosis, and its inhibition by specific siRNA, antisense or antibodies causes apoptosis in many renal cancer cells. Apoptosis induced by RLIP76 depletion is due to the inhibition of survival genes such as PI3K, ERK and Akt. Another study showed that transglutaminase 2 (TGase 2) is overexpressed in 90% of ccRCC patients. TGase 2 binds p53, inducing its transfer to autophagosome for degradation. Inhibitors such as GK921 and streptonigrin inhibit the binding between p53 and TGase 2, which has anti-cancer therapeutic potential [102,103,104]. An additional promising target for increasing the chemosensitivity of ccRCC cells is TRIM8, a crucial p53 modulator that was discussed extensively in the previous section. Indeed, the restoration of TRIM8 expression makes ccRCC and colon cancer cells sensitive to different chemotherapeutic agents [75,82].

Recently, long non-coding RNA (lncRNA) has been proposed as a new therapeutic strategy for overcoming drug resistance in genitourinary system including ccRCC, prostate, bladder and testicular cancers [105]. SRLR (sorafenib resistance-associated lncRNA) was reported to promote the resistance to multi-kinase inhibitor sorafenib [106]. In particular, SRLR has been shown to interact with the transcription factor NF-kB, which subsequently activates the expression and autocrine secretion of InterLeukin-6 (IL-6). This results in the activation of the STAT3 pathway, which interferes with the sorafenib-induced inhibition of the tyrosine kinase receptors VEGFR and PDGFR. ARSR (activated in RCC with sunitinib resistance lncRNA) significantly influences the resistance to multi-kinase inhibitor sunitinib in RCC, acting as competing endogenous RNA (ceRNA). ARSR lncRNA sequesters miR-34 (whose expression is promoted by p53) and miR-449 and therefore determines the increase of their target genes, namely AXL receptor tyrosine kinase and c-MET tyrosine kinase genes with consequent resistance to sunitinib. Interestingly, sunitinib resistance is transmitted to sensitive cells through exosome-mediated transfer [107]. NEAT1 (nuclear paraspeckle assembly transcript 1) lncRNA also shows high expression in RCC cell lines and tissues. It promotes resistance to the chemotherapy agent sorafenib by acting as a sponge for miR-34a and by interfering with the NEAT/miR-34a/c/c-MET axis [108].

It is important to deeply investigate tumor biology and the chemoresistance mechanisms linked to mutated or inactivated p53 to develop new therapeutic strategies for overcoming chemoresistance. In fact, for example, Apigenin, a plant flavonoid widely used for its anti-inflammatory action, has shown a cytotoxic effect against haematological and solid tumors [109,110], with mutant p53 inducing the apoptosis through the generation of reactive oxygen species (ROS) [111]. Another example is capsaicin, which increases the chemosensitivity of cancer cells to chemotherapeutic drugs such as cisplatin by inducing the degradation of mutated p53, restoring the activity of wild-type p53, downregulating the expression of the MDR1 gene and inducing cancer cell death [112].

## 4. Conclusions

Among cancers showing a low rate of p53 mutations and poor response to conventional therapies, RCC represents an extraordinary example of the importance of p53 pathway alterations in therapy resistance. Indeed, the fact that p53 is poorly mutated in renal carcinomas leads to an underestimation of its crucial role in the aggressiveness of the tumor and chemoresistance. Several relevant biomarkers have been proposed in RCC, but the activities of many of them are dependent on or related to p53. Moreover, the p53 wild-type status does not necessarily imply that p53 is functional, and several experimental data provide evidence of this. Therefore, it is essential to understand which pathways lead to p53 inactivation in RCC in order to develop novel therapeutic targets and improve RCC treatment.

## Figures and Tables

**Figure 1 cancers-14-05733-f001:**
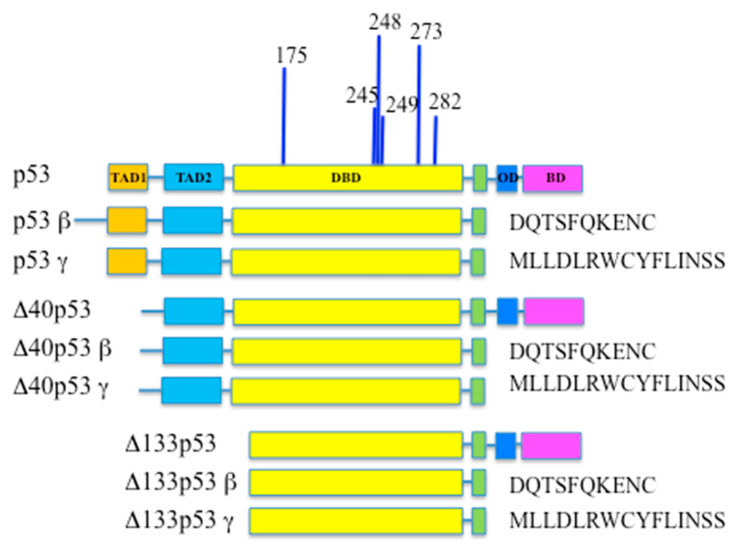
p53 tumor suppressor gene, its isoforms and functional domains: TAD1 = TransActivation Domain 1; TAD2 = TransActivation Domain 2; DBD = DNA Binding Domain; OD = Oligomerization Domain; BD = Basic Domain. The most common p53 hot spot mutation sites in cancer are reported.

**Figure 2 cancers-14-05733-f002:**
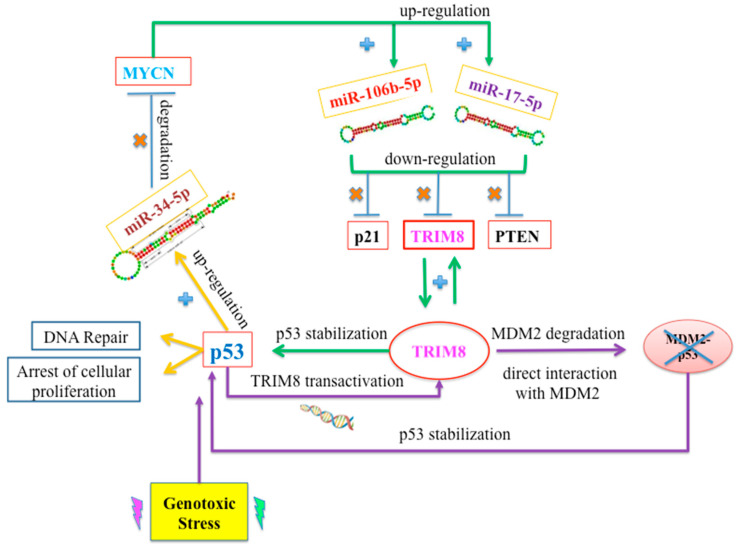
The deficit of TRIM8 in ccRCC is explained by the up-regulation of miR-17-5p and miR-106b-5p mediated by MYCN. Other crucial targets of miR-17-5p and miR-106b-5p are p21 and PTEN. By restoring the levels of TRIM8, p53 is stabilized and activates the expression of miR-34-5p, which in turn promotes the degradation of MYCN. Moreover, there is a positive feedback loop between p53 and TRIM8: Following cellular stress, p53 is stabilized, and in turn, it transactivates the expression of TRIM8. TRIM8 first mediates the degradation of MDM2 and second stabilizes p53. The stabilized p53 transactivates the expression of genes involved in DNA repair and arrest of cellular proliferation.

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
