# Peer review of "The Underestimated Role of the p53 Pathway in Renal Cancer"

_cancers, 2022, doi:10.3390/cancers14235733_

Round 1

Reviewer 1 Report

Considering that the review is thought to "evaluate the role of p53 in renal cancer" more then half of the review is a review about p53 (structure, function, regulation). The rest of the info given on p53 and renal cancer can be found already in the reviews stated below.

So my suggestion would be that the authors shorten the part on p53 and focus more on the still in part controversial role of p53 and pathways regulating it's activity and go more into depths in discussing the different publications addressing it.

Author Response

We thank the reviewer for critical reading of our manuscript and for suggestions.

We are very sorry that the added value of this review compared to the existing ones was not evident enough.

Indeed, in this review we tried to highlight that, due to the fact that p53 is poorly mutated in renal cancers, we tend to underestimate its role in chemoresistance. Indeed, there are many studies that aim to search for new biomarkers and pathways responsible for chemoresistance in RCC. This is right, but it is also true that the fact that p53 is poorly mutated, does not mean that it is functionally active and we reported the recent increasing experimental evidences to support this. Moreover, among the proposed biomarkers, a crucial role is certainly played by p53 since the cellular activities of most biomarkers identified in RCC are in some way dependent on or related to p53. Therefore, the search for novel molecular biomarkers involved in the pathways that regulate altered p53 in RCC is mandatory in order to improve early diagnosis, to evaluate the prognosis and to develop novel potential therapeutic targets for a better RCC patient’s treatment.

Anyway, we agree with the reviewer that the part describing the p53 structure, function and regulation is quite lengthy and, as suggested, we shortened this part in the revised version of the manuscript.

Furthermore we inserted a new paragraph entitled "p53 role in multi-drug resistance in RCC" in which we discussed the controversial role of p53 in the RCC chemoresistance.

Reviewer 2 Report

The manuscript entitled "Understimated Role of p53 Pathway in Renal Cancer" has been carefully reviewed regarding the niche subject selected to review by the authors. I recommend this manuscript to be published in Cancers only after some minor modifications:

1. The language needs a few revisions to correct apparent typos. (e.g. the "Understimated" in the title should be corrected to "Underestimated").

2. Despite all the efforts to prepare this review, the figures seem to have not gained enough attention from the authors. It is highly suggested to redraw them with a more scientific design and/or prettier concept telling nature.

3. In my opinion, there is a part missing from this manuscript: discussing the p53 role in multi-drug resistance as a part of the manuscript would be valuable. It is up to the authors, but that would add special scientific value to the manuscript since MDR is one of the main challenging concerns in cancer research.

Author Response

We deeply appreciate the time and effort that you spent in reviewing our manuscript and we thank you for the positive evaluation.

As suggested, the language has been revised and we apologize for any inaccuracies.

The figures were revised following the advice of the reviewer and moreover we combined figures 2 and 3 into a single figure.

As recommended by the reviewer, a new paragraph was added: "p53 role in multi-drug resistance in RCC" in which we discussed the controversial role of p53 in the chemoresistance of renal carcinomas.

Reviewer 3 Report

Overall, the review titled, “Understimated role of p53 pathway in renal cancer” by Alessandra Amendolare and team is clear, constructive and consistent. There is need revise grammar and English throughout the article. Some sentences are too long and complex.

Some of the suggestions:

In the abstract, please revise i.e. This percentage rises to about 90% (instead of 90%) if mutations and alterations in the genes that code for regulators of p53 stability and activity are considered.

 Renal Cell Carcinoma (RCC) is a clear example of cancer in which shows poor prognosis despite having a wild type p53 because of the high rate of resistance to radiotherapy or chemotherapy, which leads to recurrence, metastasis and death (please revise the sentence).

The search for novel molecular biomarkers involved in the pathways that regulate altered p53 in RCC is mandatory in order to improve early diagnosis, to evaluate the prognosis and to develop novel potential therapeutic targets for a better RCC patient’s treatment (please revise the sentence).

 between 0 and 13bp (W.S. El-Deiry WS et al., 1992); this sequence is bound by a p53 (line 138) please check why this reference is here it’s not in bibliography and does not follow journal style

Increase the font size in Figure 2 and 3.

Author Response

We deeply appreciate the time and effort that you spent in reviewing our manuscript and we thank you for the positive evaluation.

As suggested by the reviewer we completely revised the language of the manuscript, in particular of the Abstract and we apologize for any inaccuracies.

We check the references in the text following the journal style.

As suggested we increased the font size of the Figures.

Round 2

Reviewer 1 Report

I thank the authors for the effort to improve the scientific outline of the review by shortening the p53 section and adding an addition section on p53 and multi-drug resistance in RCC.